# Detection of Torquetenovirus and Redondovirus DNA in Saliva Samples from SARS-CoV-2-Positive and -Negative Subjects

**DOI:** 10.3390/v14112482

**Published:** 2022-11-09

**Authors:** Pietro Giorgio Spezia, Andreina Baj, Francesca Drago Ferrante, Sara Boutahar, Lorenzo Azzi, Angelo Genoni, Daniela Dalla Gasperina, Federica Novazzi, Francesco Dentali, Daniele Focosi, Fabrizio Maggi

**Affiliations:** 1Department of Translational Research, University of Pisa, 56126 Pisa, Italy; 2Department of Medicine and Surgery, University of Insubria, 21100 Varese, Italy; 3North-Western Tuscany Blood Bank, Pisa University Hospital, 56124 Pisa, Italy; 4Laboratory of Virology, National Institute for Infectious Diseases, Lazzaro Spallanzani IRCCS, 00161 Rome, Italy

**Keywords:** TTV, Torque Teno Virus, ReDoV, Redondovirus, saliva, SARS-CoV-2

## Abstract

Objectives: Torquetenovirus (TTV) and Redondovirus (ReDoV) are the most prevalent viruses found in the human respiratory virome in viral metagenomics studies. A large-scale epidemiological study was performed to investigate their prevalence and loads in saliva samples according to SARS-CoV-2 status. Methods: Saliva samples from 448 individuals (73% SARS-CoV-2 negative and 27% SARS-CoV-2 positive) aged 23–88 years were tested. SARS-CoV-2 and TTV were determined in saliva by specific qualitative and quantitative real-time PCRs, respectively. A sub-cohort of 377 subjects was additionally tested for the presence and load of ReDoV in saliva, and a different sub-cohort of 120 subjects for which paired saliva and plasma samples were available was tested for TTV and ReDoV viremia at the same timepoints as saliva. Results: TTV in saliva was 72% prevalent in the entire cohort, at a mean DNA load of 4.6 log copies/mL, with no difference regardless of SARS-CoV-2 status. ReDoV was found in saliva from 61% of the entire cohort and was more prevalent in the SARS-CoV-2-negative subgroup (65% vs. 52%, respectively). In saliva, the total mean load of ReDoV was very similar to the one of TTV, with a value of 4.4 log copies/mL. The mean viral loads in subjects infected with a single virus, namely, those infected with TTV or ReDoV alone, was lower than in dually infected samples, and Tukey’s multiple-comparison test showed that ReDoV single-infected samples resulted in the only true outlier (*p* = 0.004). Differently from TTV, ReDoV was not detected in any blood samples. Conclusions: This study establishes the prevalence and mean value of TTV and ReDoV in saliva samples and demonstrates the existence of differences between these two components of the human virome.

## 1. Introduction

Torquetenovirus (TTV) and Redondovirus (ReDoV) are viruses with a single, circular DNA (ssDNA) genome approximately 3000 nucleotides (nt) in length and are the most prevalent viruses found in the human respiratory virome in viral metagenomics studies [1]. Both viruses are among the simplest replication-competent and biologically successful viral entities infecting humans and are part of the vast, emerging group of ssDNA viruses associated with eukaryotic hosts [1].

However, despite these similarities, TTV and ReDoV differ widely from each other in genomic organization and genetic variability and represent the prototypes of two distinct viral families: Anelloviridae and Redondoviridae [2]. TTV is the oldest member of the Anelloviridae family (discovered in 1997), infecting most of the human population. TTV is ubiquitous across the human body and it is able to induce a chronic active infection with no associated clinical manifestation. Increasing evidence exists on the control of TTV replication exerted by the immune system, with plasma TTV loads that are higher in patients with immune dysfunctions compared to healthy controls. The relationship is more evident in subjects who undergo transient changes following perturbations in immunity (e.g., immunosuppressive therapy, transplantation, and chemotherapy [3]). Thus, PCR monitoring of TTV viremia has been recently proposed to assess the global immune function of the infected subject.

ReDoV was discovered in 2018 by metagenomic sequencing and then more widely characterized by specific-PCR investigations [4]. Differently from TTV, ReDoV does not seem to be widely distributed in the body fluids: it has been commonly detected in the respiratory tract and occasionally in the gut, while it has never been found in blood and/or other biological samples of the infected host. The virus can chronically colonize the airways [5,6], also with the simultaneous presence of genetically different strains grouped in two equally prevalent species: Vientovirus and Brisavirus [7]. The pathogenetic potential of ReDoV is still not clearly demonstrated. However, high levels of the virus are detected in patients with various clinical conditions (respiratory diseases, critical illness, autoimmune pathologies, inflammatory bowel diseases) and a particularly strong association has been demonstrated with periodontal disease [8,9,10]. According to these findings, ReDoV could be more harmful than TTV to human health.

While SARS-CoV-2-positive patients coinfected with common respiratory viruses have been widely described and the effect of a more severe disease among these patients has been established [11,12], the coinfection with TTV and ReDoV has been poorly investigated, particularly by using saliva as the biological sample.

Here, 448 saliva samples from SARS-CoV-2-negative and -positive subjects were investigated for TTV and ReDoV DNA presence and loads. The study aimed to investigate if the measurements of these viruses differed in saliva samples, if they varied in presence of SARS-CoV-2 RNA and if the measure of TTV in saliva could mirror the size of the virus circulating in the blood.

## 2. Materials and Methods

### 2.1. Study Population and Specimens

Saliva samples from 448 individuals (mean age ± SD: 53 ± 11 years, range: 23–88 years; male ratio: 1.5:1) were collected. Of these, 326 (73%) samples were from healthy individuals enrolled during the screening for SARS-CoV-2 infection, and 122 (27%) from patients admitted to the hospital after the diagnosis of COVID-19 provided by real-time RT-PCR on nasopharyngeal swabs (Aptima™ SARS-CoV-2 Assay; Hologic, San Diego, CA, USA). In both cohorts, the subjects were requested to provide saliva through the drooling technique. This technique allows for collecting only oral fluids, thus excluding mucous secretions from the oropharynx or lower respiratory tract (i.e., sputum). Saliva samples were self-collected in a sterilized 15 mL polystyrene sputum collection tube and transported to the laboratory within 2 h, where they were conserved at −80 °C for a maximum period of 6 months before use. All the 448 saliva samples were tested for SARS-CoV-2 RNA and TTV DNA presence and load. A statistically representative sub-cohort of saliva samples from 377 of 448 subjects (84%, mean age ± SD: 52 ± 11 years, range: 23–84 years; male ratio: 1.4:1) was randomly selected and investigated for ReDoV DNA detection and quantitation. The study was run after ethical approval of ASST Sette Laghi Institutional Review Board (protocol number 68/2020).

### 2.2. SARS-CoV-2 Detection

Viral nucleic acid extractions were made from 200 μL of saliva samples with the QIAamp Viral RNA mini kit (QIAGEN, Hilden, Germany) and eluted in 60 μL. Saliva specimens with high viscosity were diluted with an equivalent volume of UTM. Detection of SARS-CoV-2 was performed by RT-PCR with 5 μL of RNA template using the SARS-CoV-2 ELITe MGB^®^ Kit (ELITechGroup, Turin, Italy) which amplifies the viral RdRP, and ORF8 genes as well as the human RNase P gene as an internal control. Following the manufacturer’s instructions, a master mix was prepared to contain all components of the reaction including enzymes, nucleotides, probes, and fluorophores; 20 microliters of this mix were added per well in a 96-well plate, and 10 microliters of extracted RNA were then added. The PCR program was carried out for 45 cycles at 95 °C for 10 s, 60 °C for 30 s, and 72 °C for 20 s. Valid results were those in which the internal control gene was amplified with a Ct ≤ 35. A sample was considered positive if at least one of the genes was amplified with a Ct ≤ 40.

### 2.3. TTV DNA Detection

The presence and load of the TTV genome were determined by using a single-step TaqMan in-house real-time PCR, previously developed in our laboratories [13]. Extracted DNA was amplified with primers and probe designed on a highly conserved segment of the 5′ untranslated region of the TTV genome (sense primer 5′-GTGCCGIAGGTGAGTTTA-3′, positions 177–194; antisense primer 5′-AGCCCGGCCAGTCC-3′, positions 226–239; probe 5′-TCAAGGGGCAATTCGGGCT-3′, positions 205–223). Real-time PCR amplification was performed for 40 cycles on QuantStudio Dx real-time instrument (Thermo Fisher Scientific Inc, Waltham, MA, USA). The PCR method has high sensitivity (measuring as low as 1.1 Log TTV DNA copies/mL of saliva sample) and specificity (since it does not detect the other human anelloviruses, such as Torque Teno Mini Virus and Torque Teno Midi Virus).

### 2.4. ReDoV DNA Detection

Extracted DNA was amplified with a PCR protocol developed in our laboratories. The amplification was performed by a quantitative Taqman real-time PCR. PCR reaction was performed by Premix Ex Taq™ (Takara Bio Inc., San Jose, CA, USA) in a 25 μL volume with 5 μL of extracted DNA and 0.5 μM of each primer (sense primer: 5′-TGATGTAACATTCTATACCAAATGGA-3′, position from 1473 to 1498 nucleotide; antisense primer: 5′-ACACCTGTTTCTGATGGTACT-3′, position from 1542 to 1562 nucleotide). A Taqman probe (5’-FAM/CGAGACCAAAGGCCTCTCCCT/BHQ1-3′, position from 1501 to 1521 nucleotide) was also used at a concentration of 0.1 μM in the PCR reaction. The PCR was carried out using the following program: 95.0 °C for 1 min and then 95.0 °C for 5 s and 60.0 °C for 30 s for a total of 40 cycles. To estimate the reproducibility of the assay, a ReDoV-negative sample spiked with a known copy number of a recombinant plasmid into which the target fragment of real-time PCR was inserted, was repeated in five independent experiments, and the coefficient of variation was calculated. The reproducibility of the PCR assay was good, being the inter-assay variation lower than 0.1 Log and 1.2% in Ct. The lower detection limit was calculated to be of about 10 copies.

### 2.5. ReDoV Genetic Characterization

A segment of 1199-nt within the Rep gene of virus genome (nt 1733–2931 of the representative isolate BrisaVirus-VW, MK059759) was amplified by a qualitative single-step PCR protocol (ssPCR) by using sense primer (ReDoV_REP_F: 5′-AATTTTAGTTGGTTCGTAACAACGGT-3′, nucleotide positions 1733–1759) and antisense primer (ReDoV_REP_R2: 5′-CAAAAAATGCAAACAACAATTAAAG-3′, nucleotide positions 2907–2931). The PCR was performed on a 25 μL PCR mixture, and the amplified products was electrophoresed on agarose gel (QIAquick Gel Extraction Kit, Qiagen, Chatsworth, CA, USA) and sequenced with the Big Dye Terminator Cycle Sequencing kit (Applied Biosystems, Foster City, CA, USA) using only the ReDoV_REP_R2 primer. The sensitivity of ssPCR was previously measured by testing serial dilutions of a positive sample and was about 1000 DNA copies.

The sequences obtained were then visually verified and edited using UGENE software and a fragment of 477 base pairs was selected for phylogenetic analysis. The evolutionary history was inferred by using the Maximum Likelihood method and Tamura-Nei model and phylogenetic tree were obtained by applying Neighbor-Join and BioNJ algorithms to a matrix of pairwise distances estimated using the Tamura-Nei model. The tree is drawn to scale, with branch lengths measured in the number of nucleotide substitutions per site. Phylogenetic analyses were carried out in MEGA version 11 software (https://www.megasoftware.net/, accessed on 1 September 2022).

### 2.6. Statistical Analysis

SPSS software version 23 (IBM, Chicago, IL, USA) was used for statistical analysis. Transformed TTV and ReDoV loads in Log format were used for analysis. The Chi-square test and Fisher’s exact test were applied to evaluate the heterogeneity of contingency tables. Differences between distributions were calculated by using non-parametric the Mann–Whitney U test. The association among variables was evaluated by using the analysis of variance test, Tukey’s multiple-comparison test and the Kruskal–Wallis test. Correlations between variables were assessed using the Spearman rho correlation coefficient. All *p* values presented are based on two-tailed tests, and *p* < 0.05 was considered statistically significant.

## 3. Results

### 3.1. SARS-CoV-2 Presence in Saliva

The presence of SARS-CoV-2 RNA was investigated in all 448 saliva samples. All 122 hospitalized subjects with positive SARS-CoV-2 nasopharyngeal swabs also tested positive for SARS-CoV-2 in saliva. All 326 healthy individuals from screening cohort were negative for SARS-CoV-2 both in nasopharyngeal swab (NPS) and saliva samples. The mean SARS-CoV-2 cycle threshold (Ct) value in saliva was 30 (±5 standard deviations, SD).

### 3.2. TTV Prevalence and Loads in Saliva

The presence of TTV DNA was investigated in the entire cohort of 448 saliva samples (Table 1). The overall prevalence was 72% (325 samples), with the remaining 123 below the detection limit of the real-time PCR assay. No statistically significant difference was noted when the subjects were grouped by SARS-CoV-2 status in NPS. TTV prevalence was 73% in the SARS-CoV-2-negative group and 71% in the positive group. TTV load was precisely quantified in the saliva samples of 325 positive subjects. Total mean (±SD) was 4.6 ± 1.3 Log TTV DNA copies/mL (median: 4.7 Log; range: 1.2–7.6 Log copies/mL), with no difference regardless the SARS-CoV-2 status (Table 1). TTV load was ≤ 3.0 Log copies/mL in 12% (39/325) of TTV positive samples, and levels ≥ 6.0 Log were seen in 14% (45/325) of specimens. No correlation was found between TTV loads and age, and TTV presence and load did not differ between males and females (data not shown).

### 3.3. ReDoV Prevalence and Loads in Saliva

Overall, ReDoV DNA was investigated in 377 saliva samples (255 negative and 122 positive samples for SARS-CoV-2 infection, respectively; from Table 1), the remaining 71 having no residual aliquots for further PCRs. The overall prevalence was 61% (230 samples), significantly lower than that of TTV in the total of samples tested (*p* = 0.0004, Chi-Square test). A statistically significant difference in ReDoV prevalence was also noted when the saliva samples were grouped by SARS-CoV-2 status: ReDoV was 65% prevalent in the SARS-CoV-2-negative group and 52% in the positive group (*p* = 0.0098, Chi-Square test). When the Ct values for SARS-CoV-2 PCR detection were evaluated, no statistically significant difference was observed relative to ReDoV and/or TTV status (data not shown).

ReDoV levels were measured in 230 saliva samples. The total mean load was very similar to the one of TTV with a value of 4.4 ± 1.5 Log ReDoV DNA copies/mL (median: 4.6 Log; range: 1.2–8.0 Log copies/mL). No significant difference in ReDoV levels was observed in saliva samples with different SARS-CoV-2 status (Table 1). As shown in Figure 1, ReDoV load was ≤ 3.0 Log copies/mL in 20% (46/230) of samples, and levels ≥ 6.0 Log were seen in 13% (31/230) of specimens.

Of these 377 samples, 50% (189/377) reacted positively to both TTV and ReDoV PCRs, and 33% (124/377) were positive to only one PCR. Of these latter samples, 67% (83/124) were TTV positive and ReDoV negative, and 33% (41/124) were TTV negative and ReDoV positive. Sixty-four of 377 (17%) samples were negative for both TTV and ReDoV. Then, we stratified the viral loads of the 313 samples according to the number of viruses they harbored, and the Kruskal–Wallis analysis of variance was used. As in Figure 1, comparing the mean viral loads of single- and dual-virus infections yielded a significant heterogeneity (*p* < 0.016; Kruskal–Wallis test). The mean viral loads of the single infected samples, namely those infected with TTV or ReDoV alone, showed a lower value when compared to dually infected samples, and Tukey’s multiple-comparison test showed that ReDoV single-infected samples resulted in the only true outlier (*p* = 0.004).

### 3.4. TTV and ReDoV Loads in Simultaneous Saliva and Plasma Samples

The levels of TTV and ReDoV were also measured in simultaneous serum samples available from 120 subjects, all negative for SARS-CoV-2 infection. Sixty-nine samples tested TTV positive at copy numbers with a mean ± SD of 4.1 ± 0.6 Log copies/mL of serum (median: 4.0 Log; range: 2.8–6.1 Log copies/mL). Viral loads in plasma and saliva showed a high positive correlation in individual subjects (*r* = 0.606; *p* < 0.0001). Interestingly, 13 saliva samples tested TTV positive (total mean ± SD: 5.2 ± 1.8 Log in saliva samples), although the corresponding serum samples were below the detection threshold, while the opposite was seen in 7 saliva samples (total mean ± SD: 4.0 ± 1.0 Log in serum samples).

Finally, no serum sample was found to contain circulating ReDoV, despite the virus presence in the corresponding saliva samples.

### 3.5. Genetic Analysis of ReDoV Isolates

Sequencing was limited to 13 amplicons obtained from independent PCRs. Phylogenetic analysis showed that our isolates were related to the previously published strains and they were all classified within the Brisavirus genus (Figure 2).

## 4. Discussion

The discovery that the DNA of small, circular viruses can be found in the respiratory tract of humans [5,10,14] inspired this study on saliva samples collected from SARS-CoV-2-positive and -negative patients. Among these viruses, TTV and ReDoV deserve special attention being the most abundant components of the human respiratory virome. In the case of TTV, no evidence exists on a direct causal association with any specific human disease. Because of this and because TTV infection is highly active and prevalent among healthy people, the virus is now considered part of the normal human microflora and completely devoid of pathogenic potential [2]. On the other hand, the possibility that ReDoV plays a direct and/or indirect role in some human diseases cannot be completely excluded. Evidence exists for association between the virus and acute respiratory or other diseases, but more studies are needed for clearly demonstrating this causal relationship [8,9,10]. Then, to date, the significance of these viruses in the infected host is not fully understood and also it is unclear if their presence can influence the infection with other respiratory viruses, such as SARS-CoV-2. The study results show that while TTV and ReDoV have entirely different prevalence in plasma (close to 100% for TTV, close to 0% for ReDoV), both TTV and ReDoV are equally prevalent in saliva, with similar viral loads. While the lack of detailed clinical information about the patients studied here limits observation regarding the association of TTV and ReDoV with more severe SARS-CoV-2 diseases, the finding of particularly high prevalence and levels of both viruses in saliva suggests pursuing the observation furtherly. Although the present finding is in line with studies showing a lack of correlation between SARS-CoV-2 prognosis and TTV viral load in plasma [15,16,17], other studies have demonstrated that the measure of TTV in saliva can be useful to predict prognosis in SARS-CoV-2-positive patients. Mendes-Correa et al. revealed that TTV levels decreased over time as the symptoms of SARS-CoV-2-infected individuals resolved [18], and Merenstein et al. found that Anelloviridae showed more frequent respiratory colonization and higher titers in severe COVID-19 [19]. The relationship between ReDoV and SARS-CoV-2 is almost completely unknown, although the possibility that RedoV loads in COVID-19 patients are abnormal compared to those in healthy people has been also reported [19].

The study results extend previous published findings, confirming that high TTV levels can be revealed in saliva samples [20,21], and they give important information on the poorly explored point of ReDoV presence in saliva samples [7]. Again, the findings strengthen the concept that ReDoV infection is not systemic but mainly restricted at level of respiratory tract [5], while the concentration of TTV in saliva is significantly correlated with the load of circulating virus [22]. Despite this difference, the high prevalence, and similar loads of TTV and ReDoV found in saliva warrants further investigations on the role of circular DNA viruses as immune markers in saliva. Saliva is a biological matrix that is much more accessible than blood, and better fits the repeated sampling required for serial monitoring of immune competence, e.g., in transplant recipients. Albert et al. showed that TTV DNA was detected more frequently and at higher loads in saliva than in plasma specimens at all time points after allogeneic hematopoietic stem cell transplantation (overall, 94.2% vs. 86.5%) [23], but more studies in different populations of immunocompromised patients are definitively needed.

## 5. Limitations

These include the small sample size, lack of stratification according to COVID-19 severity, and lack of serological investigations. The latter are not commercially available for either TTV or ReDoV. A further limitation is the inability to perform correlations between SARS-CoV-2 viral loads in NPS versus saliva (the PCRs being run under different platforms, Hologic vs. ELITech).

## Figures and Tables

**Figure 1 viruses-14-02482-f001:**
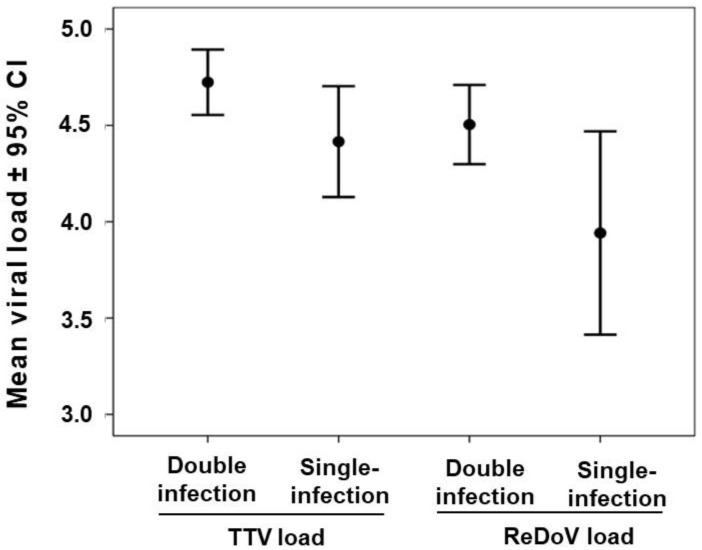
TTV and ReDoV loads of 313 saliva samples, stratified by the number of viruses carried. Viral load is expressed as the mean log DNA copies per ml of saliva ± confidence intervals (CI).

**Figure 2 viruses-14-02482-f002:**
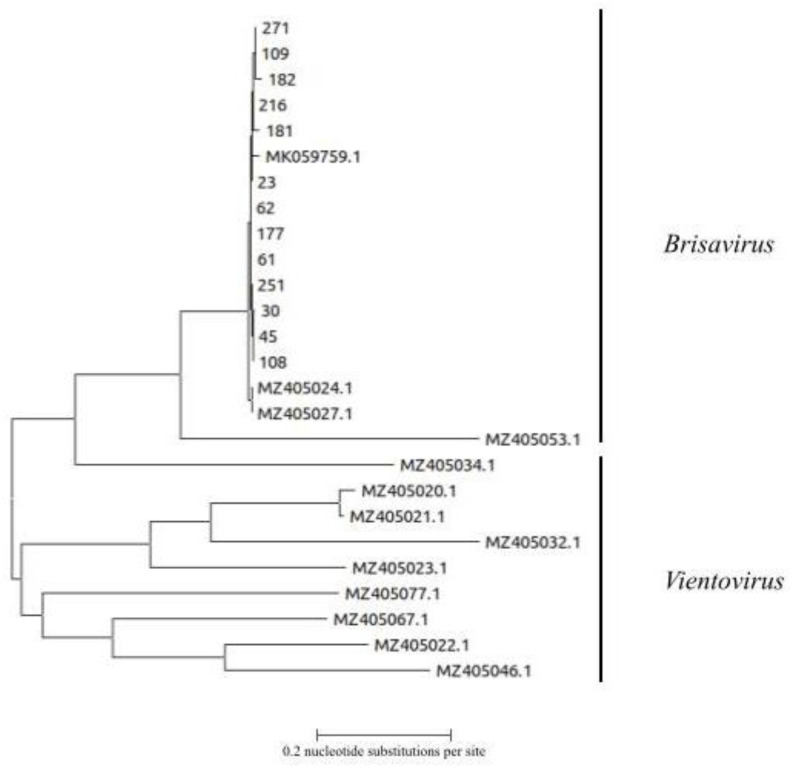
Phylogenetic analysis of 13 ReDoV sequences from the present study. The nt tree is based on a fragment of 477 base from the Rep gene of the genome viral genome. ReDoV sequences from the present study are indicated by numbers. The 13 sequences of ReDoV obtained from GenBank are indicated by accession number. Clades representing the two Redondovirus genera are indicated. The bar represents the number of substitutions per site.

**Table 1 viruses-14-02482-t001:** TTV and ReDoV prevalence and load in the saliva samples, grouped by SARS-CoV-2 RNA status.

Parameter	SARS-CoV-2 Negative	SARS-CoV-2 Positive	Total
Saliva samples examined (No.)	326	122	448
TTV DNA positive (%)	238 (73)	87 (71)	325 (72)
ReDoV DNA positive (%)	167 (65) ^a,b^	63 (52) ^c^	230 (61) ^d,e^
TTV DNA load (Log copies/mL ± 95% CI)	4.6 (4.4–4.8)	4.7 (4.3–4.8)	4.6 (4.4–4.7)
ReDoV DNA load (Log copies/mL ± 95% CI)	4.4 (4.1–4.6)	4.5 (4.0–4.7)	4.4 (4.2–4.5)

CI, confidence limit. ^a^ Value calculated on 255 of 326 (78%) SARS-CoV-2-negative saliva samples. ^b^ Statistically significant to 63 ReDoV DNA positive in SARS-CoV-2-positive group at *p* = 0.0098 (Chi-Square test). ^c^ Statistically significant to 87 TTV DNA positive in SARS-CoV-2-positive group at *p* = 0.0099 (Chi-Square test). ^d^ Value calculated on 377 of 448 (84%) saliva samples. ^e^ Statistically significant to 325 TTV DNA positive in the Total group at *p* = 0.0004 (Chi-Square test).

## Data Availability

The dataset is available from F.M. upon reasonable request.

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
