# Peer review of "Detection of Torquetenovirus and Redondovirus DNA in Saliva Samples from SARS-CoV-2-Positive and -Negative Subjects"

_viruses, 2022, doi:10.3390/v14112482_

Round 1

Reviewer 1 Report

This is a well-written and presented study evaluating TTV and ReDoV in saliva from subjects with and without SARS-CoV-2/COVID. The principal weakness is the absence of any clinical data related to COVID status. While the authors acknowledge this, it is a great pity!

Major comments:

None

Minor comments:

I have several technical questions in regard to methodology. 

1. How was the 377 cohort for ReDoV chosen?

2. What was the time between saliva collection and storage at -80C? What was the time from storage until PCR analysis?

3. Could the authors provide details about the TTV assay? This for completeness rather than just providing a reference for it [11].

4. Line 127-128. Kindly provide percentage (in addition to identifying <0.1 Log) for CV for the ReDoV Assay.

5. It would be interesting to understand how many in the screening cohort were SARS-CoV-2 positive?

6. Introduction, line 43. Remove the bracket “)”.

7. Lines 58-59: Please change: “The virus can colonize chronically the airways [5,6],…” to “The virus can chronically colonize the airways [5,6],…”

Reviewer 2 Report

Dear Authors,

Thank you for waiting for my review, I was recently unable to provide a timely review as by the date requested, for what I want to apologize, and I hope you will find my review useful for what of the major revision. It has stroked my attention with the analysis of SARS-CoV-2 patients, since The interest in TTV grew immensely after it was revealed that the number of copies in circulation was negatively correlated with the degree of immune resistance.

1.       I find the objective section missed. I would strongly urge authors to provide clearer statement of what is the aim of the study, and how this study is important from the point of view of the SARS-CoV-2 diagnostics? Was TTV used as a potential biomarker in monitoring immunological competence?

2.       What about the nomenclature? TTV is classified as Torque Teno virus, and authors provide it as the whole name, altogether. If you decide to stick with it, please use at least “Torque Teno virus” in the keywords section.

3.       Same as in the point 1 there is the issue of stating all about the TTV and RDV but not in the relation to the chosen groups. Why were the SARS-CoV-2 patients chosen for this research? The separate paragraph about it should be provided in the introduction.

4.       Please do not state the results in the description of the aim of the study (last paragraph of the introduction).

5.       Methods, lines 97 and 98: “The PCR program was carried out for 45 cycles of 95°C for 10s and 60°C for 30s, and 72°C for 20s Valid results were those in which the internal control gene was amplified with a Ct ≤ 35.” Please put the period “.” after “20s”.

6.       Could you change the Figure 1, that there is no underlining in red under the words?

7.       Discussion needs a re-do. In this way it is impossible to be provided. It is too short and is not focusing on the matters provided in the methodic. Please do as follow:

a.       Provide a separate chapter for “Limitations” from the lines 256 to 268, excluding this part from the discussion.

b.       Describe in more detail why your work had to include patients undergoing SARS-CoV-2 testing for the testing of TTV and RDV,

c.       Discuss your results with other analysis available from the saliva for TTV and RDV,

d.       How RDV and TTV are available in saliva, and what is the relation between their salivary and plasma levels?

e.       How the relation between the RDV and TTV might influence the SARS-CoV-2 infection?

In summary, as the work represents an interesting part, the aim of the study is missing, along with its best description within all the parts. Methodology is written in a good way, and as such does not need a redo, apart from up written information. The weakest part is discussion, that need more of the author’s focus. Please provide the more vast reference list.

Round 2

Reviewer 2 Report

Dear Authors,

thank you for changes in the manuscript.

Firstly, in order to strenghten the aim of the study, please develop more the section of the Discussion where you provide the informations abou the TTV and ReDoV. Provide the background of one paragraph lenght about those viruses and their relationship with the other diseases, and also their total count in the saliva.

Secondly, eliminate from the all manuscript personal references to the minimum, i.e. "our work, we provide, we show". Additionally, cancel the phrase: "Our study has several limitations." from the limitations section.

Best

Reviewer 2

Author Response

Reply to reviewer 2:

Comment 1. Firstly, in order to strenghten the aim of the study, please develop more the section of the Discussion where you provide the informations abou the TTV and ReDoV. Provide the background of one paragraph lenght about those viruses and their relationship with the other diseases, and also their total count in the saliva.

Reply and action:  As suggested, a sentence on this point has been added in the discussion section.

Comments 2. Secondly, eliminate from the all manuscript personal references to the minimum, i.e. "our work, we provide, we show". Additionally, cancel the phrase: "Our study has several limitations." from the limitations section.

Reply ancd action. Done